



**Measurement report: PM$_{2.5}$-bound nitrated aromatic compounds in Xi'an,**
**Northwest China: Seasonal variations and contributions to optical properties of**
**brown carbon**
Wei Yuan[1,6], Ru-Jin Huang[1,2], Lu Yang[1], Ting Wang[1,6], Jing Duan[1,6], Jie Guo[1], Haiyan Ni[1,7],
Yang Chen[3], Qi Chen[4], Yongjie Li[5], Ulrike Dusek[7], Colin O'Dowd[8], Thorsten Hoffmann[9]
[1]State Key Laboratory of Loess and Quaternary Geology, Center for Excellence in Quaternary
Science and Global Change, Key Laboratory of Aerosol Chemistry & Physics, Institute of Earth
Environment, Chinese Academy of Sciences, Xi'an 710061, China
[2]Institute of Global Environmental Change, Xi'an Jiaotong University, Xi'an 710049, China
[3]Chongqing Institute of Green and Intelligent Technology, Chinese Academy of Sciences,
Chongqing 400714, China
[4]State Key Joint Laboratory of Environmental Simulation and Pollution Control, College of
Environmental Sciences and Engineering, Peking University, Beijing 100871, China
[5]Department of Civil and Environmental Engineering, Faculty of Science and Technology,
University of Macau, Taipa, Macau SAR 999078, China
[6]University of Chinese Academy of Sciences, Beijing 100049, China
[7]Centre for Isotope Research (CIO), Energy and Sustainability Research Institute Groningen
(ESRIG), University of Groningen, 9747 AG, The Netherlands
[8]School of Physics and Centre for Climate and Air Pollution Studies, Ryan Institute, National
University of Ireland Galway, University Road, Galway H91CF50, Ireland
[9]Institute of Inorganic and Analytical Chemistry, Johannes Gutenberg University Mainz,
Duesbergweg 10−14, 55128 Mainz, Germany
*Correspondence to*: Ru-Jin Huang (rujin.huang@ieecas.cn)



**Abstract**
Nitrated aromatic compounds (NACs) are a group of key chromophores for brown carbon
aerosol (light absorbing organic carbon, i.e., BrC), which affects radiative forcing. The
chemical composition and sources of NACs and their contributions to BrC absorption, however,
are still not well understood. In this study, $PM_{2.5}$-bound NACs in Xi'an, Northwest China, were
investigated for 112 daily $PM_{2.5}$ filter samples from 2015 to 2016. Both the total concentrations
and contributions from individual species of NACs show distinct seasonal variations. The
seasonally averaged concentrations of NACs are 2.1 (spring), 1.1 (summer), 12.9 (fall), and
56.3 ng m$^{-3}$ (winter). Thereinto, 4-nitrophenol is the major NAC component in spring (58%).
The concentrations of 5-nitrosalicylic acid and 4-nitrophenol dominate in summer (70%), and
the concentrations of 4-nitrocatechol and 4-nitrophenol dominate in fall (58%) and winter
(55%). The NAC species show different seasonal patterns in concentrations, indicating
differences in emissions and formation pathways. Source apportionment results using positive
matrix factorization (PMF) further show large seasonal differences in the sources of NACs.
Specifically, in summer, NACs were highly influenced by secondary formation and vehicle
emissions (~80%), while in winter, biomass burning and coal combustion contributed the most
(~75%). Furthermore, the light absorption contributions of NACs to BrC are wavelength
dependent and vary greatly by seasons, with maximum contributions at ~330 nm in winter and
fall and ~320 nm in summer and spring. The differences in the contribution to light absorption
are associated with the higher mass fractions of 4-nitrocatechol ($\lambda_{max}$=345 nm) and 4-
nitrophenol ($\lambda_{max}$=310 nm) in fall and winter, 4-nitrophenol in spring, and 5-nitrosalicylic acid
($\lambda_{max}$=315 nm) and 4-nitrophenol in summer. The mean contributions of NACs to BrC light
absorption at the wavelength of 365 nm in different seasons are 0.14% (spring), 0.09%
(summer), 0.36% (fall) and 0.91% (winter), which are about 6-9 times higher than their mass
fractional contributions of carbon in total organic carbon. Our results indicate that the
composition and sources of NACs have profound impacts on the BrC light absorption.

**1 Introduction**
Brown carbon (BrC) aerosol has received growing attention over the past years, because



it can affect the atmospheric radiation balance and air quality through absorption of solar
radiation in the near ultraviolet and visible range (Feng et al., 2013; Laskin et al., 2015; Zhang
et al., 2017). Nitrated aromatic compounds (NACs) belong to a major group of BrC
chromophores. They are ubiquitous in the atmosphere and have been detected in cloud water
(Desyaterik et al., 2013), rainwater (Schummer et al., 2009), fog water (Richartz et al., 1990),
snow water (Vanni et al., 2001), as well as in gas and particle phases (Cecinato et al., 2005;
Zhang et al., 2013; Chow et al., 2015; Al-Naiema and Stone, 2017). Field studies have shown
that ~4% of BrC light absorption at 370 nm is contributed by those measured NACs (Zhang et
al., 2013; Mohr et al., 2013). In addition, with molecular structures commonly containing nitro
($-NO_2$) and hydroxyl (-OH) functional groups on the aromatic ring, NACs are harmful to human
health (Taneda et al., 2004). There is also evidence that NACs affect plant growth and
contributed to forest decline (Hinkel et al., 1989; Natangelo et al., 1999). The significant role
of NACs in the atmosphere and their adverse effects on ecosystems call for studies to
investigate their sources and characteristics.

NACs in atmospheric aerosol can be derived from primary emissions, including biomass

burning (Wang et al., 2017; Teich et al., 2017; Lin et al., 2018), coal combustion (Olson et al.,
2015; Lu et al., 2019a), and vehicle exhausts (Taneda et al., 2004; Inomata et al., 2013; Perrone
et al., 2014; Lu et al., 2019b). The emission factors of NACs from biomass burning can be over
10 mg $kg^{-1}$ (Wang et al., 2017), which makes them good tracers of biomass burning organic
aerosol (BBOA) (Hoffmann et al., 2007; Iinuma et al., 2010). Lu et al. (2019a) determined that
the emission factors of fine particulate NACs for residential coal combustion was 0.2-10.1 mg
$kg^{-1}$ and the total NAC emission from residential coal burning was nearly 200 Mg in China in
2016. NACs from vehicle exhaust also have been detected, with emission factors of up to 26.7
µg $km^{-1}$ (Lu et al., 2019b). Secondary formation from various atmospheric reactions is also an
important source of NACs. For example, photochemical oxidation of benzene, toluene (Wang
et al., 2019), and *m*-cresol (Iinuma et al., 2010) can form certain NACs. NACs can also form
in aerosol or cloud water through aqueous-phase reactions (Vione et al., 2001, 2005), for
example, photonitration of guaiacol in the aqueous phase (Kitanovski et al., 2014). However,
little is known about the importance of primary versus secondary sources for particle-bound



NACs because speciation of NACs and quantification of their sources are still very limited so
far.

Speciation of particle-bound NACs was mostly performed in Europe (Cecinato et al., 2005;

Iinuma et al., 2010; Delhomme et al., 2010; Mohr et al., 2013; Kahnt et al., 2013), and still very
scarce in Asia (Chow et al., 2015; Wang et al., 2018; Ikemori et al., 2019). In general, the
average concentrations of measured NACs vary from less than one to dozens of ng m$^{-3}$ in
different seasons and regions. As far as we know, only one study has quantified the sources of
NACs with a positive matrix factorization (PMF) receptor model (Wang et al., 2018). Here, we
carried out chemical analyses together with light absorption for PM$_{2.5}$ samples collected in
Xi'an to: 1) investigate the seasonal variations in the concentration of NACs and contributions
of individual species; 2) quantify the sources of NACs in different seasons based on PMF model;
and 3) evaluate the optical properties of NACs and their contributions to BrC light absorption.
**2 Experiments and methods**
**2.1 Aerosol sampling**

24 h-integrated PM$_{2.5}$ samples were collected in four seasons from November 2015 to

November 2016 (i.e., from 30 November to 31 December 2015 for winter; 19 April to 19 May
2016 for spring; 1 to 31 July 2016 for summer; and 9 October to 15 November 2016 for fall) in
the campus of the Institute of Earth Environment, Chinese Academy of Sciences (IEECAS,
34.22°N, 109.01°E) in Xi'an, China. The sampling site is an urban background site surrounded
by residential areas and has no obvious industrial activities. A total of 112 samples were
collected on pre-baked (780 °C, 3 h) quartz-fiber filters (20.3 × 25.4 cm, Whatman, QM-A,
Clifton, NJ, USA) by a Hi-Vol PM$_{2.5}$ sampler (Tisch, Cleveland, OH) operating at 1.05 m$^3$ min$^{-1}$.
The filter samples were stored at -20 °C until laboratory analysis.
**2.2 Chemical analysis**

The concentration of organic carbon (OC) was measured by a Thermal/Optical Carbon

Analyzer (DRI, Model 2001, Atmoslytic Inc., Calabasas, CA, USA) with the IMPROVE-A
protocol (Chow et al., 2011). Ten NACs and 19 organic markers (see Table S1) were quantified
by a gas chromatograph-mass spectrometer (GC-MS) using a well-established approach (e.g.,




Wang et al., 2006; Al-Naiema and Stone, 2017) and the details are described in Yuan et al.,
2020. Baseline separation with symmetrical peak shapes was achieved for the measured NACs
(Fig. 1). The linear ranges, instrument detection limit (IDL), instrument quantitation limit (IQL),
extraction efficiency, and regression coefficients for the measured NACs are shown in Table
S2. The response of calibration curves for the NACs was linear ($R^2 \geq 0.995$) from 10 to 5000
$\mu g\ L^{-1}$. The IDL ranged from 2 $\mu g\ L^{-1}$ to 20 $\mu g\ L^{-1}$ except for 5-nitrosalicylic acid (52.6 $\mu g$
$L^{-1}$). The IQL ranged from below 10 $\mu g\ L^{-1}$ to 70 $\mu g\ L^{-1}$ except for 5-nitrosalicylic acid (> 100
$\mu g\ L^{-1}$). The IDL and IQL are comparable to those in Al-Naiema and Stone (2017) (2.7-14.9
$\mu g\ L^{-1}$ for IDL and 8.8-49.5 $\mu g\ L^{-1}$ for IQL) and are sufficient for the quantification of our
samples.
**2.3 Light absorption of NACs**
The UV-Vis spectrophotometer equipped with a Liquid Waveguide Capillary Cell
(LWCC-3100, World Precision Instrument, Sarasota, FL, USA) was used to measure the light
absorption of methanol-soluble BrC and NAC standards, following the method established by
Hecobian et al. (2010). The absorption coefficient ($Abs_\lambda$: M $m^{-1}$) can be obtained from measured
absorption data by equation (1):
$$Abs_\lambda = (A_\lambda - A_{700})\frac{V_l}{V_a \times L}\ln(10) \qquad (1)$$
where $A_{700}$ is the absorption at 700 nm used to correct for baseline drift, $V_l$ is the volume of
methanol used for extracting the filter, $V_a$ is the volume of sampled air, L is 0.94 m for the
optical path length used in LWCC, and $\ln(10)$ is used to convert the absorption coefficient from
log base-10 to natural logarithm.
The mass absorption efficiency (MAE: $m^2\ g^{-1}$) of NAC standards in the methanol solvent
at wavelength of λ can be calculated as Laskin et al. (2015):
$$MAE_{NAC,\lambda} = \frac{A_\lambda - A_{700}}{L \times C}\ln(10) \qquad (2)$$
where C ($\mu g\ mL^{-1}$) is the concentration of the NAC standards in the methanol solvent.
The light absorption contribution of NACs to BrC at wavelength of λ ($Cont_{NAC/BrC,\lambda}$) can
be obtained using equation (3).
$$Cont_{NAC/BrC,\lambda} = \frac{MAE_{NAC,\lambda} \times C_{NAC}}{Abs_{BrC,\lambda}} \qquad (3)$$




where the $C_{NAC}$ (µg m$^{-3}$) is the atmospheric concentration of NACs and the $Abs_{BrC,\lambda}$ is the Abs
of BrC at wavelength of $\lambda$.

**2.4 Source apportionment**

The sources of NACs was resolved by PMF receptor model, which was performed by the
multilinear engine (ME-2; Paatero, 1997) through the Source Finder (SoFi) interface encoded
in Igor Wavemetrics (Canonaco et al., 2013). The input species include five to ten NACs (as
the number of NACs detected varies among seasons) and nineteen additional organic tracer
species (see Table S1). These include phthalic acid for secondary formation, picene for coal
combustion, hopanes for vehicle emission, fluoranthene, pyrene, chrysene, benzo(a)pyrene,
benzo(a)anthracene, benzo(k)fluoranthene, benzo(b)fluoranthene, benzo(ghi)perylene, and
indeno[1,2,3-cd]pyrene for combustion emission, and vanillin, vanillic acid, syringyl acetone,
and levoglucosan for biomass burning. To separate the source profiles clearly, the contribution
of those markers unrelated to a certain source was set to 0 in the respective source profile (see
Table S3).
To better understand the source origins of the NACs, air mass origins during the sampling
period were derived from backward-trajectory analysis. This method was used in trajectory
clustering based on the GIS-based software-TrajStat (Wang et al., 2009). The archived
meteorological data was obtained from the National Center for Environmental Prediction's
Global Data Assimilation System (GDAS). In this study, 72-h backward trajectories terminated
at a height of 500 m above ground level were calculated during the study period. The trajectories
were calculated every 12 h with stating times at 09:00 and 21:00 local time.

**3 Results and discussion**

**3.1 Seasonal variations of NAC composition**

The concentrations of NACs show clear seasonal differences, with the highest mean values
in winter, followed by fall, spring, and summer (see Fig. 2). The concentration ranges of total
NACs were 1.4-3.4 ng m$^{-3}$ (spring), 0.1-3.8 ng m$^{-3}$ (summer), 1.6-44.2 ng m$^{-3}$ (fall), and 20.2-
127.1 ng m$^{-3}$ (winter). The average concentrations were 2.1 ± 0.6 ng m$^{-3}$, 1.1 ± 0.8 ng m$^{-3}$, 12.9
± 11.6 ng m$^{-3}$ and 56.3 ± 23.2 ng m$^{-3}$, respectively (see Table S4). Nitrophenols (4-nitrophenol,





2-methyl-4-nitrophenol,     3-methyl-4-nitrophenol,     2,6-dimethyl-4-nitrophenol)     and
nitrocatechols (4-nitrocatechol, 3-methyl-5-nitrocatechol, 4-methyl-5-nitrocatechol) show the
highest concentrations in winter and the lowest in summer, while nitrosalicylic acids (3-
nitrosalicylic acid, 5-nitrosalicylic acid) show the highest concentrations in winter and the
lowest in spring. The average ratios between wintertime and summertime concentrations are a
factor of about 40 for nitrophenols, 175 for nitrocatechols, and 21 for nitrosalicylic acids. The
large seasonal differences in NAC concentrations might be due to the differences in sources,
emission strength and atmospheric formation processes, as discussed below. Table 1
summarizes the NAC concentrations measured in this study together with those measured in
Europe, the USA and other places in Asia. In general, the NAC concentrations in winter are
higher than those in summer, and the observed concentrations of different species are higher in
Asia than in Europe and the USA. The only exception is a study in Ljubljana, Slovenia, which
shows that in winter nitrocatechol concentrations are higher than those in Asia, likely due to
strong biomass burning activities (Kitanovski et al., 2012). The elevated concentrations of
NACs in Asia suggest that NACs may have a significant impact on regional climate and air
quality in Asia due to its optical and chemical characteristics, as discussed below.
Among all measured NACs, 4-nitrophenol, 2-methyl-4-nitrophenol, 3-methyl-4-
nitrophenol, 4-nitrocatechol and 5-nitrosalicylic acid were detected in four seasons, 3-methyl-
5-nitrocatechol and 4-methyl-5-nitrocatechol in fall and winter, 2,6-dimethyl-4-nitrophenol, 3-
nitrosalicylic acid and 4-nitro-1-naphthol only in winter, as shown in Fig. 3a. In general, 4-
nitrophenol and 4-nitrocatechol had elevated concentrations in all seasons, which is consistent
with other observations (Chow et al., 2015; Ikemori et al., 2019) and might be related to their
larger emissions or formation and longer atmospheric lifetime than other NACs (Harrison et al.,
2005; Chow et al., 2015; Finewax et al., 2018; Wang et al., 2019; Lu et al., 2019a). For example,
Lu et al. (2019a) measured the emission of NACs from coal combustion and founded that the
emission factors of 4-nitrocatechol was about 1.5-6 times higher than other NAC. Wang et al.
(2019) quantified the concentration of 4-nitrophenol and 4-nitrocatechol formed under high
$NO_x$ and anthropogenic VOC conditions, which is about 3-7 times higher than other NAC. The
concentration of 2-methyl-4-nitrophenol was higher than that of 3-methyl-4-nitrophenol in all



seasons, which is similar to previous studies (Kitanovski et al., 2012; Chow et al., 2015; Teich
et al., 2017; Ikemori et al., 2019) and likely due to the efficient formation of 2-methyl-4-
nitrophenol from photochemical oxidation of volatile organic compounds (VOCs) in the
presence of $NO_2$ (Lin et al., 2015; Wang et al., 2019). It should be noted that the contribution
of 5-nitrosalicylic acid (27%) to total NAC mass in summer is much higher than that in other
seasons (4%-13%), suggesting that 5-nitrosalicylic acid is mainly produced by secondary
formation, for example, through nitration of salicylic acid (Li et al., 2020), photochemical
oxidation of toluene in the presence of $NO_x$ (Jang and Kamens, 2001; Wang et al., 2018).
**3.2 Sources of NACs**
Correlation analysis was conducted among NACs measured in this study (Table S5). The
four nitrophenols were positively correlated with each other ($r^2$ = 0.52-0.98) and the three
nitrocatechols were also highly correlated with each other ($r^2$ = 0.94-0.96), indicating that
different nitrophenols and nitrocatechols might have similar sources or origins. Previous studies
showed that 4-nitrophenol was mainly from primary emission of biomass burning (Wang et al.,
2017), and 3-methyl-5-nitrocatechol and 4-methyl-5-nitrocatechol were identified as secondary
products from biomass burning (Iinuma et al., 2010). Positive correlations were also observed
between nitrophenols and nitrocatechols ($r^2$ = 0.59-0.90), suggesting that they were partly of
similar sources or formation processes. For example, both nitrophenols and nitrocatechols can
be emitted through biomass burning (Wang et al., 2017) and coal combustion (Lu et al., 2019a)
and can be formed by photochemical oxidation of VOCs in the presence of $NO_2$ (Wang et al.,
2019). However, for nitrosalicylic acids, the correlation between 3-nitrosalicylic acid and 5-
nitrosalicylic acid was weak ($r^2$ = 0.29). This is because 5-nitrosalicylic acid is mainly from
secondary formation by nitration of salicylic acids, while 3-nitrosalicylic acid is mainly from
combustion emission (Wang et al., 2017; Li et al., 2020). The correlations between nitrosalicylic
acids with nitrophenols ($r^2$ = 0.01-0.13) and with nitrocatechols ($r^2$ = 0.04-0.25) were also weak,
suggesting that they may have different sources or formation processes. Nitrosalicylic acids
were dominated by 5-nitrosalicylic acids, which is mainly from secondary formation
(Andreozzi et al., 2006; Wang et al., 2018). On the other hand, nitrophenols and nitrocatechols
were dominated by 4-nitrophenol and 4-nitrocatechol, respectively, which are mainly from



primary emissions (Wang et al., 2017; Lu et al., 2019a).

To identify and quantify the sources of NACs observed in Xi'an, the PMF model was

employed and four major factors were resolved with uncertainties < 15%. The factor profiles
are shown in Fig. S1. The first factor, vehicle emission, characterized by high levels of hopanes,
shows large relative contributions to NACs in spring and summer. Direct traffic emissions of
NACs have also been verified in laboratory studies (Tremp et al., 1993; Perrone et al., 2014).
The second factor is considered to be coal combustion for residential heating and cooking,
which is characterized with the higher loadings of picene, benzo(a)pyrene,
benzo(b)fluoranthene, benzo(k)fluoranthene, indeno[1,2,3-cd]pyrene, and
benzo(ghi)perylene. This factor accounted for ~40% of the NACs in winter. The emission of
NACs from coal combustion for residential usage was reported by Lu et al. (2019a), which
showed emission factors of 0.2 to 10.1 mg kg$^{-1}$. It is worth noting that with the emission control
of residential coal burning after 2017, the contribution feature of coal burning to NACs could
be different. The third source is identified as secondary formation because of the highest level
of phthalic acid and its highest contribution in summer. The formation of secondary NACs is
also supported by both field and modeling studies (Harrison et al., 2005; Iinuma et al., 2010;
Yuan et al., 2016). The last source factor, with high loadings of levoglucosan, vanillic acid,
vanillin and syringyl acetone, was identified as biomass burning, which has higher
contributions in fall and winter. The emission of NACs from biomass burning was reported by
field studies, and was considered to be an important source of NACs (Mohr et al., 2013; Lin et
al., 2016; Teich et al., 2017).

The sources contributions for NACs in Xi'an are shown in Fig. 4, which shows obvious

seasonal differences. In spring, vehicular emission (41%) was the main contributor to NACs.
Secondary formation (26%) and biomass burning (20%) also contributed significantly. In
summer, secondary formation had the highest contribution (45%), which was likely due to
enhanced photochemical oxidation leading to the formation of NACs. Besides, vehicular
emission also contributed significantly (34%) in summer. In fall, biomass burning (45%)
contributed the most, while secondary formation (30%) and vehicular emission (23%) also had
significant contributions. In winter, coal burning (39%) and biomass burning (36%) were the





main contributors, which can be attributed to emissions from residential heating activities. It is
worth noting that the absolute concentrations of NACs attributed by vehicle emission (see Table
S6) were higher in winter than those in spring and summer, yet these differences of less than
20 times are not as significant as the differences (spring and summer vs. winter) for NACs
attributed by other primary emissions (> 80 times for coal burning and > 40 for biomass
burning). These results indicate that anthropogenic primary sources are the main contributors
to NACs in Xi'an. Secondary formation also contributes significantly to NACs, especially in
summer. Further comprehensive field studies are necessary for understanding the formation
mechanisms of NACs under different atmospheric conditions.
**3.3 Backward trajectory analysis of NACs**

To reveal the source origins of the NACs, the concentrations of NACs were grouped

according to their trajectory clusters that represent different air mass origins, as shown in Fig.
5. In general, the air masses from local emissions (Cluster 1 in spring and fall and Cluster 2 in
summer and winter), which showed the features of small-scale and short-distance air transport,
caused significant increases in NAC concentrations. As for regional transport, the air masses
from the neighboring Gansu province across Baoji city before arriving at Xi'an presented
higher concentrations of NACs in fall and winter (Cluster 2 and Cluster 3, respectively). In
addition, air masses from Xinjiang across Gansu caused increased concentrations of NACs in
spring and summer (Cluster 2 and Cluster 1, respectively). A small proportion of air masses
from the northwest (Cluster 3 in spring and Cluster 1 in winter), the south (Cluster 3 in summer)
and the west (Cluster 3 in fall), which showed long or moderate transport patterns, are related
to the lowest concentrations of NACs. This may be due to the long-distance transport or
relatively clean air from those regions. In the same season, the source origins of air masses were
different between clusters, thus causing the difference in concentrations of NACs. However,
the composition of NACs was similar between clusters, which is comparable to the results of
Chow et al. (2015).
**3.4 Light absorption of NACs**

The correlations between NAC concentration and $Abs_{BrC,365}$ for each season are shown in





Fig. S2. The correlations are stronger in fall ($r^2$ = 0.68) and winter ($r^2$ = 0.63) compared to those
in spring ($r^2$ = 0.15) and summer ($r^2$ = 0.40). These results indicate that NACs are important
components of BrC chromophores in fall and winter.
Fig. 6 shows the contributions of NACs to BrC light absorption at wavelength from 300
to 500 nm ($Abs_{BrC,300-500}$) as well as the carbon mass contributions of NACs to OC. The
contributions of NACs to $Abs_{BrC,300-500}$ are wavelength dependent and vary largely in different
seasons. High contributions at wavelengths of 350-400 nm were observed in fall and winter,
but the contributions in spring and summer were mainly at wavelengths shorter than 350 nm.
These results may be due to the high proportion of nitrocatechols in fall and winter (see
discussion above), which have strong light absorption at wavelength above 350 nm (see Fig.
S3). The seasonal average contributions of NACs to $Abs_{BrC,365}$ were highest in winter (0.91 ±
0.30%), followed by fall (0.36 ± 0.22%), spring (0.14 ± 0.04%), and summer (0.09 ± 0.06%)
(see Table S4). These contributions were comparable to a previous study where eight NACs
were measured (Teich et al., 2017). The contributions of NACs to $Abs_{BrC,365}$ in winter were
about 10 times higher compared to those in summer, which could be due to the high emissions
of NACs in winter. Alternatively, enhanced atmospheric oxidizing capacity in the summer can
lead to enhanced formation of secondary NACs or the degradation/bleaching of certain NACs
(Barsotti et al., 2017; Hems and Abbatt, 2018; Wang et al., 2019) which might eventually reduce
the contributions in summer. The fractions of NACs to total OC also show obvious seasonal
variation, with average contributions higher in winter (0.14 ± 0.05%) and fall (0.05 ± 0.02%)
and lower in spring (0.02 ± 0.01%) and summer (0.01 ± 0.01%). The contributions of NACs to
BrC light absorption at 365 nm are, however, 6-9 times larger than their carbon mass
contributions to total OC. Our results echo previous studies that even small amounts of
chromophores can have a non-negligible impact on the optical characteristics of BrC due to
their disproportional absorption contributions (Mohr et al., 2013; Zhang et al., 2013; Teich et
al., 2017; Xie et al., 2017).
The daily contributions of the individual NACs to light absorption of total NACs at
wavelength of 300-500 nm are shown in Fig. 7. Similar to the concentration fractions in NACs,
nitrocatechols were the main contributors in winter and fall with contributions of 38-65% and





18-62%, respectively. On the other hand, nitrophenols dominated in spring and summer with
contributions of 61-96% and 27-100%, respectively. As for nitrophenols, 4-nitrophenol was the
most important chromophore, followed by 2-methyl-4-nitrophenol, 3-methyl-4-nitrophenol,
and 2,6-dimethyl-4-nitrophenol (only observed in winter). As for nitrocatechols, 4-
nitrocatechol was the main contributor in four seasons, while 3-methyl-5-nitrocatechol and 4-
methyl-5-nitrocatechol also contributed significantly in fall and winter. For nitrosalicylic acids,
5-nitrosalicylic acid contributed in all four seasons but contributed the most in summer, while
3-nitrosalicylic acid was only observed in winter, which could be attributed to their different
sources, as discussed above.

The seasonal contributions of individual NACs to total light absorption of NACs at

wavelength of 365 nm are shown in Fig. 3b. The relative contribution trends of 4-nitrophenol >
4-nitrocatechol > 2-methyl-4-nitrophenol > 5-nitrosalicylic acid > 3-methyl-4-nitrophenol, 4-
nitrophenol > 4-nitrocatechol > 5-nitrosalicylic acid > 2-methyl-4-nitrophenol > 3-methyl-4-
nitrophenol, 4-nitrocatechol > 4-nitrophenol > 4-methyl-5-nitrocatechol >3-methyl-5-
nitrocatechol > 5-nitrosalicylic acid > 2-methyl-4-nitrophenol > 3-methyl-4-nitrophenol, 4-
nitrocatechol > 4-methyl-5-nitrocatechol > 4-nitrophenol > 3-methyl-5-nitrocatechol > 2-
methyl-4-nitrophenol > 4-nitro-1-naphthol > 5-nitrosalicylic acid > 3-methyl-4-nitrophenol >
3-nitrosalicylic acid > 2,6-dimethyl-4-nitrophenol were observed in spring, summer, fall and
winter, respectively. These trends were different from their concentration fractions in OC,
which may be mainly due to the differences in light absorption ability (see Fig. S3). These
results suggest that mere compositional information of NACs might not be directly translated
into impacts on optical property, because they have startlingly different absorption properties.
**4 Conclusion**

In this study, ten individual NAC species were quantified, together with 19 organic

markers, in $PM_{2.5}$ in Xi'an, Northwest China. The average concentrations of NACs were 2.1,
1.1, 12.9, and 56.3 ng $m^{-3}$ in spring, summer, fall, and winter, respectively. Higher
concentrations of NACs in winter than in summer were also observed in previous studies in
Asia, Europe and the USA. Four major sources of NACs were identified in Xi'an based on
PMF analysis, including vehicle emission, coal combustion, secondary formation and biomass



burning. On average, in spring, vehicular emission (41%) was the main contributor of NACs,
and secondary formation (26%) and biomass burning (20%) also had relatively large
contributions. In summer, secondary formation contributed the most (45%), which was likely
due to the enhanced photochemical formation of secondary NACs that outweights photo-
degradation/bleaching. Besides, vehicular emission (34%) also had significantly contribution
in summer. In fall, biomass burning (45%) contributed the most, and secondary formation (30%)
and vehicular emission (23%) also made significant contributions. In winter, coal burning (39%)
and biomass burning (36%) contributed the most, which can be attributed to emissions from
residential heating activities. Backward trajectory cluster analyses indicate that both regional
and local contributions for NACs were significant in Xi'an. Local contributions were 53, 47,
66 and 44% in the four seasons, and regional transport was mainly through the northwest
transport channel. The light absorption contributions of NACs to BrC were quantified and also
showed large seasonal variations. The seasonal average contributions of total NACs to BrC
light absorption at wavelength of 365 nm ranged from 0.1% to 0.9%, which were 6-9 times
higher than their carbon mass fractions in total OC. Our results suggest that even a small amount
of chromophores can have significant impacts on the optical characteristics of BrC and more
studies are needed to better understand the seasonal differences in chemical composition and
formation processes of NACs and the link with their optical properties.

*Data availability.* Raw data used in this study are archived at the Institute of Earth Environment,
Chinese Academy of Sciences, and are available on request by contacting the corresponding
author.
*Supplement.* The Supplement related to this article is available online at
*Author contributions.* RJH designed the study. Data analysis was done by WY, LY, and RJH.
WY, LY and RJH interpreted data, prepared the display items and wrote the manuscript. All
authors commented on and discussed the manuscript.
*Competing interests.* The authors declare that they have no conflict of interest.






*Acknowledgements*. This work was supported by the National Natural Science Foundation of China (NSFC) under Grant No. 41877408, 41925015, 91644219, and 41675120, the Chinese Academy of Sciences (no. ZDBS-LY-DQC001, XDB40030202), the National Key Research and Development Program of China (No. 2017YFC0212701), and the Cross Innovative Team fund from the State Key Laboratory of Loess and Quaternary Geology (No. SKLLQGTD1801). Yongjie Li would like to acknowledge financial support from the Multi-Year Research grant (MYRG2017-00044-FST and MYRG2018-00006-FST) from the University of Macau.

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





**Table 1.** Mean and standard deviation (if applicable) of the measured mass concentrations of
individual NAC in Xi'an in comparison to those in other studies.

| Locations | Concentrations (ng m$^{-3}$) | | | | | | | | | | Reference |
|---|---|---|---|---|---|---|---|---|---|---|---|
| | 4NP | 2M4NP | 3M4NP | 2,6DM4NP | 4N1N | 4NC | 3M5NC | 4M5NC | 3NSA | 5NSA | |
| **Europe** | | | | | | | | | | | |
| TROPOS, Germany, winter 2014 | 7.09 | 3.64 | 2.60 | 0.65 | | | | | 1.36 | 0.94 | Teich et al., 2017 |
| | (7.08) | (3.05) | (2.22) | (0.58) | | | | | (1.02) | (0.75) | |
| Melpitz, Germany, summer 2014 | 0.06 | 0.04 | 0.03 | | | | | | 0.17 | 0.09 | Teich et al., 2017 |
| | (0.03) | (0.00) | (0.00) | | | | | | (0.15) | (0.09) | |
| Melpitz, Germany, winter 2014 | 4.09 | 3.64 | 2.44 | 0.91 | | | | | 0.66 | 0.32 | Teich et al., 2017 |
| | (3.27) | (3.06) | (2.20) | (0.90) | | | | | (0.69) | (0.24) | |
| Ljubljana, Slovenia, summer 2010 | 0.15 | 0.05 | <0.03 | | | 0.24 | 0.1 | 0.06 | 0.09 | 0.18 | Kitanovski et al., 2012 |
| Ljubljana, Slovenia, winter 2010 | 1.8 | 0.75 | 0.61 | | | 75 | 34 | 29 | 1.3 | 1.4 | Kitanovski et al., 2012 |
| Villa Ada park, Rome, spring 2003 | 17.8 | | 7.8 | 5.9 | | | | | | | Cecinato et al., 2005 |
| | (5.6) | | (2.6) | (2.9) | | | | | | | |
| Waldstein, Germany, summer 2014 | | | | | | | | | 0.17 | 0.23 | Teich et al., 2017 |
| | | | | | | | | | (0.11) | (0.12) | |
| **USA** | | | | | | | | | | | |
| Research Triangle Park, USA, summer 2013 | 0.018 | 0.005 | | | | 0.057 | | | | | Xie et al., 2019 |
| | (0.027) | (0.009) | | | | (0.042) | | | | | |
| Lowa City, USA, fall 2015 | 0.63 | 0.08 | | | | 1.60 | | 1.61 | | 0.14 | Al-Naiema and Stone, 2017 |
| | (0.48) | (0.05) | | | | (2.88) | | (1.77) | | (0.08) | |
| **Asia** | | | | | | | | | | | |
| Hong Kong, China, spring 2012 | 0.36 | 0.18 | 0.03 | 0.01 | | 0.25 | 0.05 | 0.05 | | | Chow et al., 2015 |
| Hong Kong, China, summer 2012 | 0.54 | 0.3 | 0.02 | 0.01 | | 1.48 | 0.63 | 0.25 | | | Chow et al., 2015 |
| Hong Kong, China, fall 2012 | 0.92 | 0.39 | 0.04 | 0.01 | | 2.45 | 0.94 | 0.44 | | | Chow et al., 2015 |
| Hong Kong, China, winter 2012 | 1.13 | 0.65 | 0.07 | 0.01 | | 2.39 | 1.35 | 0.53 | | | Chow et al., 2015 |
| Xianghe, China, summer 2013 | 0.98 | 0.32 | 0.09 | 0.06 | | | | | 1.21 | 0.88 | Teich et al., 2017 |
| | (0.78) | (0.21) | (0.07) | (0.05) | | | | | (1.45) | (0.64) | |
| Wangdu, China, summer 2014 | 2.63 | 0.68 | 0.21 | 0.06 | | | | | 3.14 | 1.63 | Teich et al., 2017 |
| | (2.66) | (0.78) | (0.35) | (0.09) | | | | | (3.05) | (0.78) | |
| Xi'an, China, spring 2016 | 1.19 | 0.24 | 0.18 | | | 0.28 | | | | 0.15 | This study |
| | (0.36) | (0.08) | (0.05) | | | (0.18) | | | | (0.15) | |
| Xi'an, China, summer 2016 | 0.45 | 0.10 | 0.07 | | | 0.16 | | | | 0.29 | This study |
| | (0.28) | (0.10) | (0.06) | | | (0.11) | | | | (0.41) | |
| Xi'an, China, fall 2016 | 3.6 | 0.73 | 0.44 | | | 3.9 | 1.23 | 1.35 | | 1.72 | This study |
| | (2.6) | (0.54) | (0.35) | | | (4.0) | (1.34) | (1.24) | | (2.3) | |
| Xi'an, China, winter 2015 | 15.6 | 4.5 | 3.4 | 0.55 | 1.16 | 15.5 | 6.4 | 6.2 | 0.84 | 2.3 | This study |
| | (6.6) | (1.72) | (1.52) | (0.39) | (0.53) | (7.4) | (3.7) | (2.9) | (0.56) | (2.4) | |
| Nagoya, summer 2013 | 1.1 | 0.49 | 0.17 | | 0.98 | 0.74 | | 0.081 | 0.33 | 0.75 | Ikemori et al., 2019 |
| | (0.54) | (0.48) | (0.13) | | (1.5) | (0.72) | | (0.077) | (0.38) | (0.84) | |
| Nagoya, Japan, fall 2013 | 7.0 | 3.2 | 1.1 | | 0.76 | 6.8 | | 1.6 | 0.27 | 0.67 | Ikemori et al., 2019 |
| | (3.9) | (2.7) | (0.76) | | (0.64) | (10.8) | | (2.9) | (0.20) | (0.41) | |

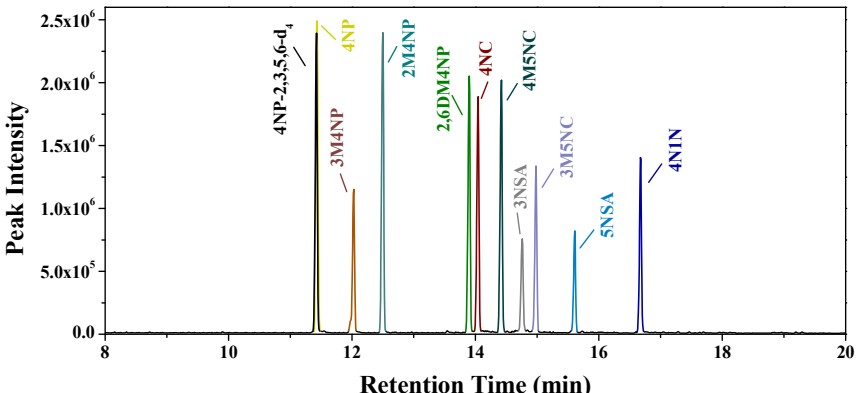


**Figure 1.** Selected ion monitoring chromatograms for the nitrated aromatic compound standards (2 ug mL$^{-1}$). (4NP-2,3,5,6-d$_4$: 4-nitrophenol-2,3,5,6-d$_4$, 4NP: 4-nitrophenol, 3M4NP: 3-methyl-4-nitrophenol, 2M4NP: 2-methyl-4-nitrophenol, 2,6DM4NP: 2,6-dimethyl-4-nitrophenol, 4NC: 4-nitrocatechol, 4M5NC: 4-methyl-5-nitrocatechol, 3NSA: 3-nitrosalicylic acid, 3M5NC: 3-methyl-5-nitrocatechol, 5NSA: 5-nitrosalicylic acid, 4N1N: 4-nitro-1-naphthol).





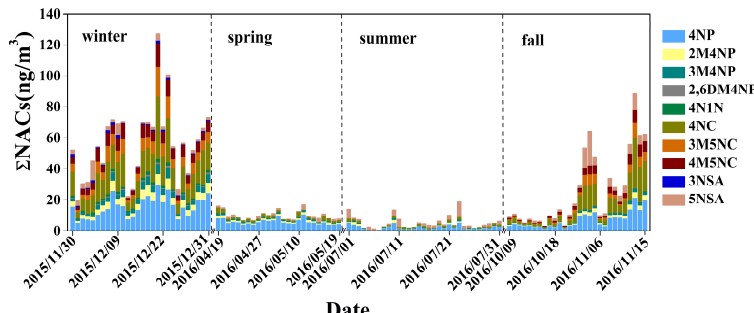

**Figure 2.** Time series of the concentrations of nitrated aromatic compounds in the aerosol

sample (spring and summer × 5, fall ×2). The full names of these compounds are shown in Table

S1.



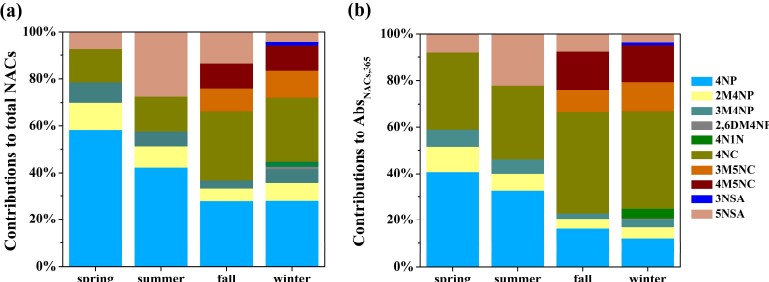

**Figure 3.** The average contributions of individual nitrated aromatic compounds to (a) the total concentration and (b) the total light absorption at wavelength 365 nm of particulate nitrated aromatic compounds in four seasons. The full names of these compounds are shown in Table S1.



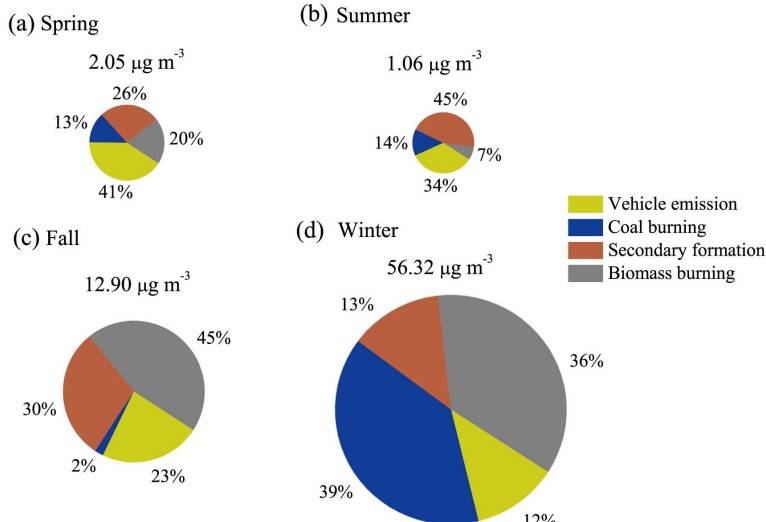

**Figure 4.** Contributions of source factors to the concentrations of NACs in four seasons.



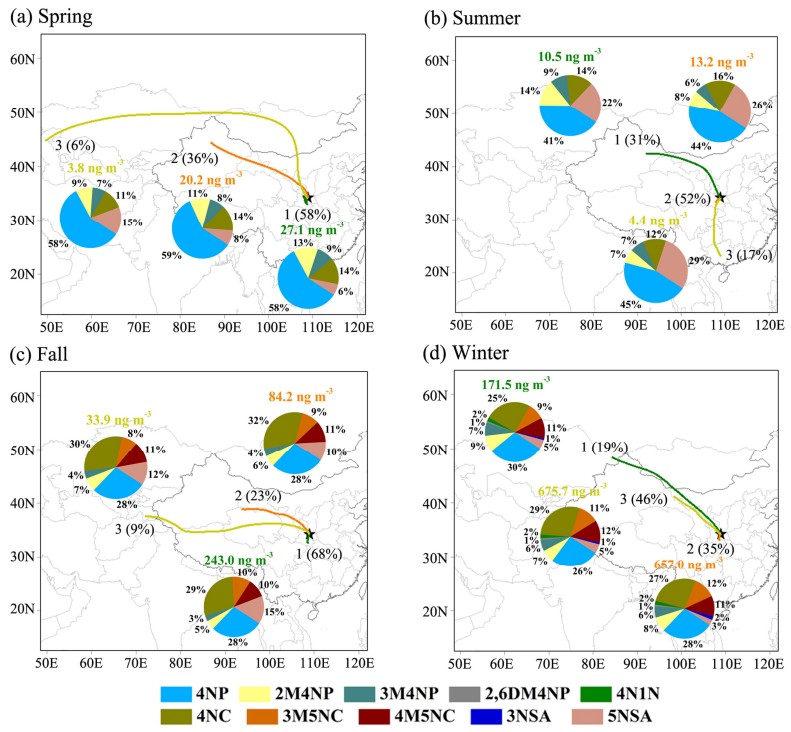

**Figure 5.** NACs at each 72-h backward trajectory cluster during (a) spring, (b) summer, (c) fall

and (d) winter. The full names of these compounds are shown in Table S1.





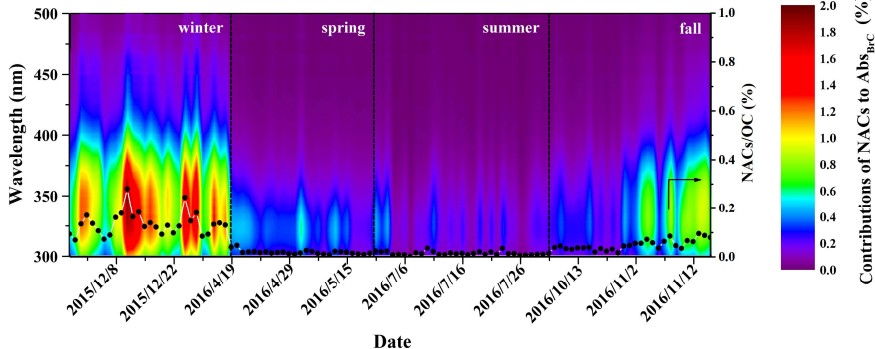

589

**Figure 6.** Time series of the light absorption contributions of total NACs to Abs of brown carbon over the wavelength from 300 to 500 nm (color scale and left axis), and the ratio of concentration of NACs to organic carbon (dots and right axis).





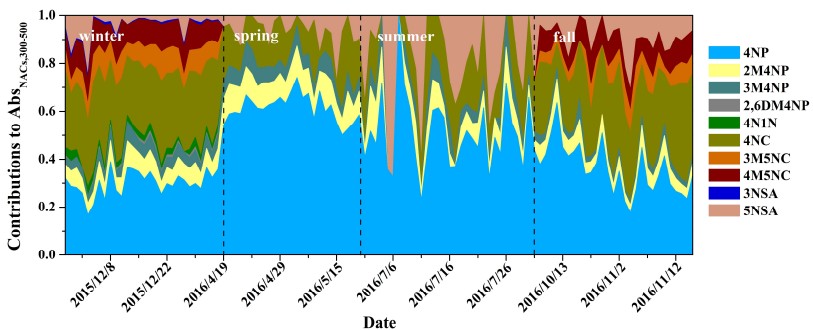

590

**Figure 7.** Daily contributions of individual NACs to light absorption of total NACs at

wavelength of 300-500 nm. The full names of these compounds are shown in Table S1.