# Peer review of "Measurement report: PM2.5-bound nitrated aromatic compounds in Xi'an,"

_Atmospheric Chemistry and Physics, 2020_

## Referee Comment (RC1) · Anonymous Referee #3 · 11 Oct 2020

This work provides a comprehensive report to the variation, sources, origins, and light absorption of nitrated aromatic compounds (NACs) in four seasons in a mega city in Northwest China. The results highlight the elevated concentrations and enhanced light absorption of NACs in winter in East Asia and confirm the dominant contributions from combustions sources including coal combustion, biomass burning, and vehicle exhausts. The manuscript is generally well written with clear logic, full discussion, and fluent language. It can be accepted after addressing a few minor comments.

Specific comments:

[Figure]

1. Line 105-107, are any blank samples collected or obtained in this study?

2. Line 118-119, is any inner standard used when determining the concentrations of NACs?

3. What is the wavelength range of the UV-Vis spectrophotometer used in this study?

4. Line 262-263, it's better to derive some implications to pollution control here.

5. Line 271-275, were the relatively high concentrations of NACs in the air masses from Gansu and Xinjiang mainly caused by the intensive emissions from urban areas along the trajectories?

6. Line 332-333, the difference in the light absorption ability among different NACs is of course the major cause. Suggest elaborating on the differences here, e.g., 4NC has high light absorption ability.

---

## Referee Comment (RC2) · Anonymous Referee #2 · 19 Oct 2020

The paper reports the contribution of PM2.5-bound nitrated aromatic compounds to the optical properties of brown carbon. Seasonal variations of concerned species were discussed as well. The topic is interesting and suitable for the journal, and the paper is well organized and understandable. However, some problems need further discussion. In conclusion, I suggest it for publication after the authors addressing the following specific points:

1. Line 63-65, considering that this paper mainly focuses on the optical properties of NACs, the introduction should include more previous research findings regarding the

light absorption ability of NACs rather than just one sentence.

2. Line 66-67, could the author add a few more sentences on why NACs are harmful to human health?

3. Line 153, what are the uncertainties of the input species?

4. Line 154, The constrain of specific species in different sources will influence the Q value of the solution, thus the setting should be extremely cautious. After this kind of setting, is the %dQ value acceptable? Is the PMF solution still robust?

5. Line 160, any reason to choose a 72-h backward trajectory instead of 24-h or 48h?

6. Line 298-300, besides the high emissions of NACs in winter, are any other PM2.5 components that may contribute to the enhanced light absorption between 300-500nm during winter?

7. Section 3.4, based on the PMF results, the author may consider using multilinear regression analysis to investigate which source contributes most to the light absorption ability of NACs.

---

## Author Comment (AC1) · 7 Dec 2020

The authors thank the referees to review our manuscript and particularly for the valuable comments and suggestions that have significantly improved the manuscript. We provide below point-by-point responses (in blue) to the referees' comments and have made changes accordingly in the revised manuscript.

Referee #2

The paper reports the contribution of $PM_{2.5}$-bound nitrated aromatic compounds to the optical properties of brown carbon. Seasonal variations of concerned species were discussed as well. The topic is interesting and suitable for the journal, and the paper is well organized and understandable. However, some problems need further discussion.

In conclusion, I suggest it for publication after the authors addressing the following specific points:

1. Line 63-65, considering that this paper mainly focuses on the optical properties of NACs, the introduction should include more previous research findings regarding the light absorption ability of NACs rather than just one sentence.

Response: Thanks for pointing this out. We have added more previous research findings in line 66-73, it now reads "…; Teich et al., 2017; Li et al., 2020). For example, Zhang et al. (2013) estimated the contribution of NACs to BrC light absorption of ~4% in the Los Angeles Basin. Mohr et al. (2013) calculated the contribution of NACs to BrC light absorption of about 4% in Detling, United Kingdom. Teich et al. (2017) investigated the contribution of NACs to BrC light absorption during six campaigns of 0.02-4.41% for acidic conditions and 0.02-9.86% for alkaline conditions. Li et al. (2020) estimated the contribution of NACs to BrC light absorption in Beijing of 0.28-3.44% in fall and 1.03-6.49% in winter."

2. Line 66-67, could the author add a few more sentences on why NACs are harmful to human health?

Response: We have added a few more sentences in line 75-77, it now reads "…For

example, NACs can interact with DNA and cause mutagenesis (Purohit and Basu, 2000; Ju and Parales, 2010). NACs can also damage cells, resulting in cell degeneration and canceration (Kovacic and Somanathan, 2014)."

3. Line 153, what are the uncertainties of the input species?

Response: We re-checked the uncertainties (RSDs) and have now added these values in the revised manuscript. In line 160, it now reads "..., with uncertainties (RSD) < 10%."

4. Line 154, The constrain of specific species in different sources will influence the Q value of the solution, thus the setting should be extremely cautious. After this kind of setting, is the %dQ value acceptable? Is the PMF solution still robust?

Response: The constrain of specific species in different sources do affect the Q value of the solution. In this study, the $Q/Q_{exp}$ value was 1-5 after setting, which is acceptable, and the PMF was ran in the robust model.

5. Line 160, any reason to choose a 72-h backward trajectory instead of 24-h or 48h?

Response: We have added the reason in Line 171-172, it now reads "According to the lifetimes of different secondary species (Wojcik and Chang, 1997; Chow et al., 2015), ..."

6. Line 298-300, besides the high emissions of NACs in winter, are any other PM2.5 components that may contribute to the enhanced light absorption between 300-500 nm during winter?

Response: In addition to NACs, some PAHs also have strong light absorption capacity in wavelength of 300-500 nm (Huang et al., 2018; Lin et al., 2018). In winter, the emissions of PAHs increase because of heating activities, which may contribute to the enhanced light absorption.

7. Section 3.4, based on the PMF results, the author may consider using multilinear

regression analysis to investigate which source contributes most to the light absorption ability of NACs.

Response: Multilinear regression analysis and PMF receptor model can both be used to investigate the sources of BrC. In this study, to get non-negative result, we used PMF rather than multilinear regression analysis to analyse the sources of NACs, and the results are discussed in Section 3.2. "Sources of NACs".

References

Chow, K. S., Huang, X. H. H., and Yu, J. Z.: Quantification of nitroaromatic compounds in atmospheric fine particulate matter in Hong Kong over 3 years: field measurement evidence for secondary formation derived from biomass burning emissions, Environ. Chem., 13, 665–673, doi:10.1071/EN15174, 2015.

Huang, R. J., Yang, L., Cao, J., Chen, Y., Chen, Q., Li, Y., Duan, J., Zhu, C., Dai, W., Wang, K., Lin, C., Ni, H., Corbin, J. C., Wu, Y., Zhang, R., Tie, X., Hoffmann, T., O'Dowd, C., and Dusek, U.: Brown carbon aerosol in urban Xi'an, northwest China: the composition and light absorption properties, Environ. Sci. Technol., 52, 6825-6833, doi:10.1021/acs.est.8b02386, 2018.

Ju, K.-S. and Parales, R. E.: Nitroaromatic compounds, from synthesis to biodegradation, Microbiol. Mol. Biol. Rev., 74, 250-272, 2010.

Kovacic, P. and Somanathan, R.: Nitroaromatic compounds: environmental toxicity, carcinogenicity, mutagenicity, therapy and mechanism, J. Appl. Toxicol., 34, 810-824, 2014.

Li, X., Yang, Y., Liu, S., Zhao, Q., Wang, G., and Wang, Y.: Light absorption properties of brown carbon (BrC) in autumn and winter in Beijing: composition, formation and contribution of nitrated aromatic compounds, Atmos. Environ., 223, 117289, 2020.

Lin, P., Fleming, L. T., Nizkorodov, S. A., Laskin, J., and Laskin, A.: Comprehensive molecular characterization of atmospheric brown carbon by high resolution mass spectrometry with electrospray and atmospheric pressure photoionization, Anal. Chem., 90, 12493–12502, 2018.

Mohr, C., Lopez-Hilfiker, F. D., Zotter, P., Prévôt, A. S., Xu, L., Ng, N. L., Herndon, S. C., Williams, L. R., Franklin, J. P., Zahniser, M. S., Worsnop, D. R., Knighton, W. B., Aiken, A. C., Gorkowski, K. J., Dubey, M. K., Allan, J. D., and Thornton, J. A.: Contribution of nitrated phenols to wood burning brown carbon light absorption in Detling, United Kingdom during winter time, Environ. Sci. Technol., 47, 6316–6324, https://doi.org/10.1021/es400683v, 2013.

Purohit, V. and Basu, A. K.: Mutagenicity of nitroaromatic compounds, Chem. Res. Toxicol., 13, 673–692, 2000.

Teich, M., van Pinxteren, D., Wang, M., Kecorius, S., Wang, Z., Müller, T., Mocnik, G., and Herrmann, H.: Contributions of nitrated aromatic compounds to the light absorption of water-soluble and particulate brown carbon in different atmospheric environments in Germany and China, Atmos. Chem. Phys., 17, 1653–1672, https://doi.org/10.5194/acp-17-1653-2017, 2017.

Wojcik, G. S. and Chang, J. S.: A re-evaluation of sulfur budgets, lifetimes, and scavenging ratios for eastern north America, J. Atmos. Chem., 26, 109-145, 1997.

Zhang, X. L., Lin, Y. H., Surratt, J. D., and Weber, R. J.: Sources, composition and absorption angstrom exponent of light-absorbing organic components in aerosol extracts from the Los Angeles Basin, Environ. Sci. Technol., 47, 3685–3693, doi:10.1021/es305047b, 2013.

Referee #3

This work provides a comprehensive report to the variation, sources, origins, and light absorption of nitrated aromatic compounds (NACs) in four seasons in a mega city in Northwest China. The results highlight the elevated concentrations and enhanced light absorption of NACs in winter in East Asia and confirm the dominant contributions from combustions sources including coal combustion, biomass burning, and vehicle exhausts. The manuscript is generally well written with clear logic, full discussion, and fluent language. It can be accepted after addressing a few minor comments.

Specific comments:

1. Line 105-107, are any blank samples collected or obtained in this study?

Response: In this study, at least one blank filter sample was measured for light absorption and organic compounds for every ten ambient samples. In line 126-127, it now reads, "At least one blank filter sample was measured for every ten ambient samples."

2. Line 118-119, is any inner standard used when determining the concentrations of NACs?

Response: In our study, 4-nitrophenol-2,3,5,6-d$_4$ was used as an internal standard to correct for potential loss for NAC quantification (Chow et al., 2015).

3. What is the wavelength range of the UV-Vis spectrophotometer used in this study?

Response: 300-700 nm.

4. Line 262-263, it's better to derive some implications to pollution control here.

Response: Thanks for pointing this out. In line 279-280, it now reads "…, suggesting that control of anthropogenic emissions (biomass burning and coal burning) is important for mitigating pollution of NACs in this region."

5. Line 271-275, were the relatively high concentrations of NACs in the air masses from Gansu and Xinjiang mainly caused by the intensive emissions from urban areas along the trajectories?

Response: As far as we known, there was no studies reporting the concentration of NACs in Gansu and Xinjiang. However, the annual average concentration of $PM_{2.5}$ was about 40 μg/m$^3$ in 14 cities in Gansu in 2016, especially, the $PM_{2.5}$ concentration in Lanzhou in 2016 was over 50 μg/m$^3$ (Liao et al., 2020). The annual average concentration of $PM_{2.5}$ was about 55 μg/m$^3$ in 16 cities in Xinjiang in 2016 (Rupakheti et al., 2021). Therefore, it is possible that NACs were transported/formed along the trajectories from areas with strong emissions.

6. Line 332-333, the difference in the light absorption ability among different NACs is of course the major cause. Suggest elaborating on the differences here, e.g., 4NC has high light absorption ability.

Response: Thanks for pointing this out. In line 353-354, it now reads "...For example, 4-nitrocatechol has lower mass concentration, but higher light absorption contribution, compared to 4-nitrophenol."

References

Chow, K. S., Huang, X. H. H., and Yu, J. Z.: Quantification of nitroaromatic compounds in atmospheric fine particulate matter in Hong Kong over 3 years: field measurement evidence for secondary formation derived from biomass burning emissions, Environ. Chem., 13, 665–673, doi:10.1071/EN15174, 2015.

Liao, Q., Jin, W., Tao, Y., Qu, J., Li, Y., and Niu, Y.: Health and economic loss assessment of $PM_{2.5}$ pollution during 2015–2017 in Gansu province, China, Int. J. Environ. Res. Public Health, 17, 3253, 2020.

Rupakheti, D., Yin, X., Rupakheti, M., Zhang, Q., Li, P., Rai, M., and Kang, S.: Spatio-temporal characteristics of air pollutants over Xinjiang, northwestern China, Environ. Pollut., 268, 115907, 2021.

---

## Author Response (AR2)

Dear Prof. Willy,

Thank you very much for your time, and particularly for your careful reading and valuable suggestion. We have made all changes accordingly. Thank you again!

For the Main text:

Line 45: Replace "by seasons" by "by season".

Response: Change made.

Line 66: Replace "by those" by "by the".

Response: Change made.

Lines 67 and 71: Replace "Li et" by "X. Li et".

Response: Change made.

Line 88: Replace "combustion was" by "combustion were".

Response: Change made.

Line 100: Replace "still very" by "is still very".

Response: Change made.

Line 107: Replace "on PMF" by "on the PMF".

Response: Change made.

Line 126: Replace "et al., 2020" by "et al. (2020)" and replace "was measured" by "was analyzed".

Response: Change made.

Lines 132, 134, 181-183, 360, 632 (within Figure 4), and 633 (within Figure 5): Several concentration data are given with too many significant figures; two significant

figures normally suffice and three suffice when the first significant figure is "1"; thus, as examples, replace "52.6" by "53", "127.1" by "127", "56.3" by "56", "12.90" by "12.9", and "243.0" by "240".

Response: Change made.

Line 137: Replace "The UV-Vis" by "A UV-Vis".

Response: Change made.

Line 141: Replace "measured" by "the measured".

Response: Change made.

Line 145: Replace "in LWC" by "in the LWC".

Response: Change made.

Line 157: Replace "of NACs was resolved by PMF measured" by "of the NACs were resolved by the PMF".

Response: Change made.

Line 172: Replace "of different" by "of the different".

Response: Change made.

Line 175: Replace "with stating" by "with starting".

Response: Change made.

Line 179: Replace "of NACs" by "of the NACs".

Response: Change made.

Line 199: Replace "to its" by "to their".

Response: Change made.

Line 209: Replace "founded that the emission factors of 4-nitrocatechol was" by "found that the emission factors of 4-nitrocatechol were".

Response: Change made.

Line 210: Replace "than other NAC" by "than those of other NACs" and replace "concentration of" by "concentrations of".

Response: Change made.

Line 212: Replace "which is about 3-7 times higher than other NAC" by "and found that they are about 3-7 times higher than those of other NACs".

Response: Change made.

Line 218: Replace "than that in" by "than in".

Response: Change made.

Line 220: Replace "(Li et al., 2020), photochemical" by "(M. Li et al., 2020) and photochemical".

Response: Change made.

Line 223: Replace "among NACs" by "among the NACs".

Response: Change made.

Line 226: Replace "different" by "the different".

Response: Change made.

Lines 237-238: Replace "Li et al." by "M. Li et al.".

Response: Change made.

Line 238: Replace "correlations between" by "correlations of".

Response: Change made.

Line 241: Replace "which is mainly" by "which are mainly".

Response: Change made.

Line 245: Replace "of NACs" by "of the NACs".

Response: Change made.

Line 251: Replace "characterized with" by "characterized by".

Response: Change made.

Line 265: Replace "sources contributions" by "source contributions".

Response: Change made.

Line 285: Replace "of NACs" by "of the NACs".

Response: Change made.

Line 315: Replace "contributions were" by "contributions are".

Response: Change made.

Line 336: Replace "in four seasons" by "in all four seasons".

Response: Change made.

Line 347: Replace ", 4-nitrocatechol" by ", and 4-nitrocatechol".

Response: Change made.

Line 351: Replace "trends were" by "trends are".

Response: Change made.

Line 357: Replace "Conclusion" by "Conclusions".

Response: Change made.

Line 359: Replace "of NACs" by "of the NACs".

Response: Change made.

Line 368: Replace "had significantly" by "had a significant".

Response: Change made.

Line 380: Replace "BrC and more" by "BrC. More".

Response: Change made.

Line 429: Replace "Atmospheric Pollution Research," by "Atmos. Pollut. Res., 1,".

Response: Change made.

Line 510: Replace "Environmental Research" by "Environ. Res.".

Response: Change made.

Line 543: Replace "Total. Environ." by "Total Environ.".

Response: Change made.

Line 615: Replace "individual NAC" by "individual NACs".

Response: Change made.

Lines 625, 630, 634 and 638: Replace "these compounds are shown" by "the compounds are given".

Response: Change made.

Line 628: Replace "The average" by "Average".

Response: Change made.

For the Supplement:

Line 33: Replace "in bracket" by "in parentheses".

Response: Change made.

Line 37: Replace "sources contributions" by "source contributions" and within Table S6,
replace twice "Sources contribution" by "Source contribution".

Response: Change made.

Line 47: Replace "NAC standard" by "NAC standards".

Response: Change made.